# The Molecular Characterization of Virulence Determinants and Antibiotic Resistance Patterns in Human Bacterial Uropathogens

**DOI:** 10.3390/antibiotics11040516

**Published:** 2022-04-13

**Authors:** Naveed Ahmed, Hira Khalid, Mariam Mushtaq, Sakeenabi Basha, Ali A. Rabaan, Mohammed Garout, Muhammad A. Halwani, Abbas Al Mutair, Saad Alhumaid, Zainab Al Alawi, Chan Yean Yean

**Affiliations:** 1Department of Medical Microbiology and Parasitology, School of Medical Sciences, Universiti Sains Malaysia, Kubang Kerian, Kelantan 16150, Malaysia; namalik288@gmail.com; 2Department of Microbiology, Faculty of Life Sciences, University of Central Punjab, Lahore 54000, Punjab, Pakistan; 3Department of Medical Education, King Edward Medical University, Lahore 54000, Punjab, Pakistan; hirakhalid6001@gmail.com (H.K.); mariammushtaq105@gmail.com (M.M.); 4Department of Community Dentistry, Faculty of Dentistry, Taif University, P.O. Box 11099, Taif 21944, Saudi Arabia; sakeena@tudent.edu.sa; 5Molecular Diagnostic Laboratory, Johns Hopkins Aramco Healthcare, Dhahran 31311, Saudi Arabia; 6College of Medicine, Alfaisal University, Riyadh 11533, Saudi Arabia; 7Department of Public Health and Nutrition, The University of Haripur, Haripur 22610, Khyber Pakhtunkhwa, Pakistan; 8Department of Community Medicine and Health Care for Pilgrims, Faculty of Medicine, Umm Al-Qura University, Makkah 21955, Saudi Arabia; magarout@uqu.edu.sa; 9Department of Medical Microbiology, Faculty of Medicine, Al Baha University, Al Baha 4781, Saudi Arabia; mhalwani@bu.edu.sa; 10Research Center, Almoosa Specialist Hospital, Al-Ahsa 36342, Saudi Arabia; abbas.almutair@almoosahospital.com.sa; 11College of Nursing, Princess Norah Bint Abdulrahman University, Riyadh 11564, Saudi Arabia; 12School of Nursing, Wollongong University, Wollongong, NSW 2522, Australia; 13Administration of Pharmaceutical Care, Al-Ahsa Health Cluster, Ministry of Health, Al-Ahsa 31982, Saudi Arabia; saalhumaid@moh.gov.sa; 14Division of Allergy and Immunology, College of Medicine, King Faisal University, Al-Ahsa 31982, Saudi Arabia; zalalwi@kfu.edu.sa

**Keywords:** MRSA, XDR, antimicrobial resistance, antibiotic stewardship, virulence, UTIs

## Abstract

The high rates of bacterial infections affect the economy worldwide by contributing to the increase in morbidity and treatment costs. The present cross-sectional study was carried out to evaluate the prevalence of bacterial infection in urinary tract infection (UTI) patients and to evaluate the antimicrobial resistance rate (AMR) in a Tertiary Care Hospital in Lahore, Pakistan. The study was conducted for the period of one year from January 2020 to December 2020. A total of 1899 different clinical samples were collected and examined for bacterial cultures using standard procedures. Samples were inoculated on different culture media to isolate bacterial isolates and for identification and susceptibility testing. A total of 1107/1899 clinical samples were positive for *Staphylococcus aureus* (*S. aureus*), *Pseudomonas aeruginosa (P. aeruginosa)*, *Escherichia coli (E. coli)* and other bacterial isolates. Methicillin-resistant *S. aureus* (MRSA) prevalence was 16.93% from these positive cases. MRSA strains were found to be highly resistant to amikacin, clindamycin, fusidic acid, gentamicin and tobramycin, while highest sensitivity was noted against vancomycin (100%) and linezolid (100%). MRSA and high rates of multidrug resistance (MDR) pose a serious therapeutic burden to critically ill patients. A systematic and concerted effort is essential to rapidly identify high-risk patients and to reduce the burden of AMR.

## 1. Introduction

Urinary tract infections (UTIs) are among the most common infectious diseases of humans. UTIs commonly result from bacteria, generally Gram-negative microorganisms, such as *Acinetobacter species* (spp.), *Proteus* spp., *P. aeruginosa*, *Escherichia coli* and *Klebsiella* spp. [1,2]. Among the Gram-positive bacteria, *S. aureus*, coagulase-negative *staphylococci* (CONS) and *Enterococcus* spp. are common bacteria causing UTIs [3]. The resistant lines are responsible for an excessive mortality rate of approximately 25,000 Europeans yearly [4], and problems related to UTIs contribute to this high mortality rate [5].

Some of the bacterial species can survive in both aerobic and anaerobic conditions. Invasion, colonization and mediation of host defense subversion are all characteristics of key virulence factors implicated in the pathophysiology of UTIs [6,7]. In addition, the pathogenicity islands (PAIs) sometimes include cryptic or functional genes encoding mobility factors such as integrases, transposons and insertion sequence elements, which is evidence of their mobility and may encourage and contribute to the emergence spread of AMR [8,9].

Fimbriae, particularly type 1 and P fimbriae, are present on bacterial cell surfaces as virulence factors [10]. These fimbriae aid in host cell attachment, tissue invasion, biofilm development and the production of cytokines. Flagella, capsular lipopolysaccharides and outer membrane proteins are examples of bacterial cell surface virulence factors. Secreted virulence factors include hemolysin and siderophores [11]. These virulence factors are crucial in allowing bacteria to invade the urinary tract and survive despite a well-functioning host defense system. The virulence factors such as adhesive subunit of type 1 fimbriae (*fimH*), *usp*, *AIM* and *IMP* could also play an essential role in AMR mechanisms among different bacteria [1,5].

UTIs are usually treated with broad-spectrum antibiotics. The irrelevant use of many antibacterial drugs has resulted in the development of exclusive resistance rates against various antibiotics in different bacteria throughout the world, leading to the emergence of MDR lines of different bacterial pathogens [12,13]. The rate of AMR and the excessive usage of many antibiotics vary extensively amongst different countries [14,15]. According to the European Survey of Antibiotic Consumption, resistant strains are responsible for a high mortality rate in European countries each year, with UTI complications accounting for a significant portion of this increased mortality [16]. The Infectious Diseases Society of America advises conducting regional surveillance to track changes in uropathogenic sensitivity in particular places [15].

The rising prevalence of MDR bacterial strains, particularly in developing countries, leads to the overuse of broad-spectrum antibiotics, including fluoroquinolones, cephalosporins and aminoglycosides, which drives up treatment and hospitalization costs. The AMR among Gram-negative bacteria is on the rise in many countries [4]. Among uropathogens, extended spectrum beta lactamases (ESBLs) have emerged as a significant mechanism of AMR [17]. Bacteria that produce β-lactamase enzymes have the ability to hydrolyze the core β-lactam structural ring in β-lactam-containing antibiotics. This ring deactivates the bactericidal properties of antibiotics [18]. 

UTIs are a significant burden in Pakistan as well [5]. Bacteria that cause hospital-acquired infections (HAIs) have acquired resistance to various antibacterial drugs, highlighting the need for community-wide AMR surveillance [18]. Furthermore, the presence of both virulence determinants and drug resistance genes simultaneously might have an additive influence on the severity of infections [1]. Keeping in view the current situation, the present study was conducted to profile the antibiotic susceptibility pattern of various uropathogens and to determine the molecular characterization of genes demonstrating virulence with antibiotic resistance among identified bacterial isolates.

## 2. Results

### 2.1. Collection of Samples

A total of 1899 samples were collected from the suspected patients, but the symptomatic UTIs were not differentiated from asymptomatic bacteriuria. From these 1899 samples, 1534 were urine, 94 were Foley catheter tips, 104 were cystoscopic urine and 167 were nephrostomy urine. Each sample was collected in a tightly capped sterile container. The inappropriate/mislabeled samples were rejected and sent back to the respective department for fresh sample collections. After microscopic examination of wet smears from urine samples (*n* = 1805), a total of 1071 samples showed the presence pus cells and organisms (bacteria).

### 2.2. Bacterial Growth and Colony Identification

After the inoculation and incubation period, Petri dishes were evaluated for bacterial growth. The *S. aureus* colonies (small white and yellow colonies) were found on blood agar plates. Growth of *E. coli* (mucoid lactose fermenter pink colonies) was found on cysteine electrolyte deficient (CLED) agar and MacConkey agar plates, while growth of *P. aeruginosa* was found on both CLED and MacConkey agar plates.

A total of 1107 samples were found to be positive for different bacteria as described in Table 1, including 681 *E. coli* isolates, 142 *P. aeruginosa*, 41 *K. pneumoniae*, 7 *Proteus mirabilis*, 19 *Acinetobacter baumanni*, 6 *Enterobacter cloacae*, 6 *Citrobacter* spp., 183 *S. aureus*, 4 *Enterococcus faecium* and 18 *Enterococcus faecalis* bacterial isolates.

From these 1107 bacterial isolates, 1006 isolates of *E. coli*, *P. aeruginosa* and *S. aureus* were selected for further susceptibility profiling and the molecular identification of specific genes. 

### 2.3. Isolation and Identification of Pseudomonas aeruginosa

*P. aeruginosa* growth on blood agar showed large irregular β-hemolytic colonies, while on the MacConkey agar, the colonies were colorless and non-lactose fermented (NLF). Gram-negative rods were seen in the microscopic examination after Gram staining. The oxidase test was performed for *P. aeruginosa* identification, and it was found to be positive. Based on colonial morphology, different biochemical tests and blood hemolysis, 142 isolates were selected for further processing, *n* = 95 from urine, *n* = 15 from cystoscopic urine, *n* = 24 from nephrostomy urine and *n* = 8 from Foley tip samples.

### 2.4. Bacterial Identification by Different Biochemical Tests

*S. aureus* showed positive reactions for the catalase test, while *Streptococcus* showed negative reactions. The further identification test specifically for *S. aureus* was the coagulase and DNA test. Both reactions were found to be positive. All *E. coli* isolates were found to be positive for the indole test while showing negative reactions against citrate test. *P. aeruginosa* showed the positive reactions for the oxidase test. The final identification using API kits was confirmed after 24 h of incubation.

### 2.5. Bacterial Identification by Gram Staining

*S. aureus* appears as Gram-positive cocci in Gram staining, while *E. coli* and *P. aeruginosa* appear as Gram-negative rods.

### 2.6. Antibiotic Susceptibility Assay

Once the bacterial isolates were identified and isolated, they were subjected to antibiotic susceptibility assays/testing (AST) using the gold standard disk diffusion method.

#### Interpretation of Antibiotic Susceptibility Pattern

The antibiotic susceptibility results of all bacterial strains are collectively interpreted in Table 2 and Table 3. The AMR rates of each bacterial isolate against specific antibiotics are shown in percentages.

### 2.7. Molecular Identification

From the total of 681 *E. coli* isolates, 593 (87.07%) were found to be positive for *fimH*, and 576 (84.58%) for the *usp* gene, and from 142 *P. aeruginosa* isolates, 67 (47.18%) were found to be positive for *IMP* and 48 (33.80%) for the *AIM* gene, while all of the 31 MRSA isolates were positive for both *nuc* and *mecA* genes.

## 3. Discussion

*P. aeruginosa* is a very well-known pathogen responsible for HAIs in hospitals, particularly in intensive care units (ICUs) [14,15]. It can cause various infections in immunocompromised patients [3]. According to a previous study conducted on hospitalized cancer patients, the prevalence of bacterial infections caused by *P. aeruginosa* was 55.36% [1]. In the current study, 142 *P. aeruginosa* isolates were found based on colonial morphology, *n* = 95 from urine, *n* = 15 from cystoscopic urine, *n* = 24 from nephrostomy urine and *n* = 8 from Foley tip samples. A high resistance to aztreonam (83.09%) among *P. aeruginosa* isolates was found, while maximum sensitivity was found against Colistin (96.48%).

One of the previous studies reported that *hlyA*, *cnf1*, *iroN*, *pap*, *iuc*, *afa*, *ompT*, *iha* and *irp2* virulence results were 50.4, 50.4, 42.27, 50.4, 10.56, 8.13, 4.87, 17.88 and 11.38%, respectively. Significant correlation (*p* < 0.05) was seen between *hlyA*, *cnf1*, *pap with afa* and *ompT*. Results demonstrated the high significance of virulence and antibiotic resistance in patients with UTI in Iran [17]. In another previous study, *bla-CTX-M* was the most predominant of the all-inclusive range β-lactamases (ESBL), while *bla-TEM* was the second most prevalent gene [19]. There has been a vast discrepancy found among UTIs because of generally isolated bacteria and those resulting from particularly less commonly isolated microorganism [3,20]. In the present study, 67.69% of isolated pathogens were *E. coli*, from which 87.07% of isolates were found to be positive for *fimH* and 84.58% for *usp* gene, consistent with the results of a previous study conducted in Lahore, Pakistan [5].

A previous study conducted in Iran [17] found bacterial growth in the following prevalence rates: *E. coli* 12.21%, *Klebsiella* spp. 8.39%, *Enterobacter cloaeca* 6.1%, *Proteus* spp. 5.34%, *P. aeruginosa* 15.26%, *S. epidermidis* 8.39%, *S. saprophyticus* 6.1%, *S. xylosus* 3.81% and *Viridance streptococci* 7.63%. The MRSA cases in Pakistan are known to be high, and to date many examinations have been performed to study the disease transmission of distinctive MRSA clones in Pakistan [18,20,21]. In a previous study from Pakistan, a total of 44 MRSA isolates were isolated from two tertiary care hospitals [21]. However, results of the current study showed that 16.93% of *S. aureus* (MRSA) isolates were resistant to oxacillin (cefoxitin), while no cases were found for vancomycin-resistant *S. aureus* (VRSA).

MRSA causes pneumonia and septicemia because of its exceptionally destructive potential and the expanding articulation of hereditary determinants of AMR [18]. A previous study performed in Portugal reported that MRSA was related to 44% of HAIs, and the death rate was around 20%. The authors found that 49% of isolates were MRSA, and 51% were MSSA with no VRSA strain. All of the MRSA isolates were tested for the amplification of *mecA* gene and were found to be positive [16]. 

Results of the present study showed that the *P. aeruginosa* isolates were more resistant to aztreonam (83.09%) and cefepime (48.59%), while the isolates showed maximum sensitivity to colistin (96.48%), polymyxin B (95.78%) and tazobactum (90.85%). The *E. coli* isolates showed high resistance to nalidixic acid (94.71%), ampicillin (93.24%) and amoxicillin (88.83%). The prevalence rate of MRSA was 16.93%, from different clinical samples, while the other 83.07% of the isolates were MSSA. The bacterial identification was also tested by molecular identification of antibiotic resistance in *E. coli*, *P. aeruginosa* and *S. aureus* isolates. From the total 681 *E. coli* isolates, 593 (87.07%) were found to be positive for *fimH* and 576 (84.58%) for the *usp* gene, and from 142 *P. aeruginosa* isolates, 67 (47.18%) were found to be positive for *IMP* and 48 (33.80%) for the *AIM* gene. The findings of this study could elucidate the coexistence of virulence genes and high AMR rates in various uropathogens.

## 4. Materials and Methods

### 4.1. Collection of Samples

All of the samples were collected from the kidney setting of a tertiary care hospital in Lahore, Pakistan over the period of 1 year (January 2020 to December 2020). The patients from the outpatient department (OPD) and those who were admitted to the inpatient department (IPD) were selected for the current study. Clinical samples of Foley catheter tips (*n* = 94), cystoscopy urine (*n* = 104), nephrostomy urine (*n* = 167) and midstream clean-catch urine (*n* = 1534) were collected.

### 4.2. Sample Processing for Bacterial Cultures

After the collection, all of the samples were transported to the main laboratory for further processing. For the isolation and identification of bacterial pathogens, the samples were inoculated on different culture media. CLED agar (Thermo Fisher Scientific Inc. Waltham, United States) was used to inoculate urine samples, while chocolate agar, blood agar and MacConkey agar were used for inoculation of samples other than urine. After inoculation, the Petri dishes were initially incubated at a temperature of 37 °C for 24 h [22]. After the incubation period, the isolated bacteria were identified by physical colonial characteristics and different biochemical identification tests [23]. Finally, after identifying bacterial isolates, all of the isolates were subjected to AST [24]. 

Patients with asymptomatic bacteriuria have no such sign and symptoms that can be attributed to bacteria in the urine. This is completely different from the evaluation and management of symptomatic bacteriuria or UTIs. Diagnosis of asymptomatic bacteriuria can be made using urine cultures. Either a properly collected clean-catch specimen or a catheterized specimen could be acceptable. According to the Infectious Diseases Society of America (IDSA), a colony count of 10^5^ colony forming units (CFU)/mL for one bacterial species from the voided urine is considered an active bacterial infection. This is also the case for catheterized specimens [25,26]. 

### 4.3. Wet Smear Microscopic Examination of Urine Samples

Before inoculation of urine samples on agar media, a wet smear examination was performed to determine the presence of pus cells and organisms/bacteria. Urine samples (5 mL) were centrifuged at 3500 RPM for 2 min, and the supernatant was discarded. From the sediment, 1–2 drops were taken on a clean sterilized glass slide, and a coverslip was placed. The slide was observed under a 40X lens of a compound microscope (Olympus, Tokyo, Japan). The presence of >4 pus cells and organisms was considered an indication of the possibility of any bacterial infections. The smears were also assessed for the presence of yeast or candida [26,27].

### 4.4. Microscopic Identification of Bacterial Isolates

Pure colonies were obtained from the culture plates and mixed with a drop of distilled water (dH2O) on a sterilized glass slide to prepare a thin smear. The slide was air-dried and subjected to Gram staining [28].

### 4.5. Biochemical Identification of Isolates

After the 24 h incubation at a temperature of 37 °C, the bacterial colonies were then identified according to the colonial morphology characteristics, Gram staining results and the biochemical profile for an individual bacterial isolate. The biochemical tests were performed with each pure bacterial colony. Tests were performed as per the standard procedures for bacterial identification [28].

The expected *S. aureus* colonies were first tested for catalase (performed on a sterile glass slide) to differentiate *Staphylococcus* from *Streptococcus* [29]. The further identification test specifically for *S. aureus* was the coagulase (performed on disposable reaction cards) [30] and DNA test (performed on MH agar Petri dishes) [31]. All lactose fermenter isolates were tested for the indole and citrate test. The expected *P. aeruginosa* isolates were tested for the oxidase test [32].

Analytical profile index (API) kits were used for the final identification of the *S. aureus* and Gram-negative bacterial isolates (Biomeriux, Marcy-l’Etoile, France). In order to avoid risk of contamination, the test was performed in a Class II, type A2 biosafety cabinet (Bioevopeak, Jinan, China). The pure bacterial colonies were picked with a sterilized wire loop and mixed with sterile 5 mL suspension medium (0.85% NaCl) in a screw cap tube. From this tube, the mixture was then poured into each box/well of the kit. After inoculation of API kit, it was incubated at 37 °C for 24 h [20,33]. After 24 h results were recorded, the kit was again incubated for the next 24 h and the final results were noted after 48 h using the API website (https://apiweb.biomerieux.com/login) (accessed from 1 January 2020 to 31 December 2020). Then, the pure isolated bacterial colonies were stored in glycerol broth at −80 °C.

### 4.6. Antibiotic Susceptibility Testing (AST)

After the final isolation and identification of bacterial isolates, they were subjected to the AST using the disk method according to the standard operating procedures as provided by CLSI guidelines (2019) [24].

The McFarland standards were used to standardize the concentration of inoculum. It was prepared with 1% hydrogen sulfide (H_2_SO_4_) (Merck, Darmstadt, Germany) and 1.175% barium chloride (BaCl_2_) (Merck, Darmstadt, Germany) to produce different concentrations of 0.5 McFarland standards. After the preparation, it was stored at 4 °C to be used further for AST [5].

Using the disk diffusion method, the AST was performed according to the CLSI guidelines (2019). ATCC Strain-25923 of *S. aureus* was used as a control. On a single Petri dish, a maximum of 6 antibiotic disks were placed and then incubated at 37 °C for 24 h [10].

### 4.7. Antibiotic Panel for Enterobacteriaceae

The following antibiotics (Thermo Fisher Scientific Inc., Washington, USA) were used for antibiotic susceptibility testing of *Enterobacteriaceae*:

Ampicillin (AMP). amp-clavulanic acid (AMC), amikacin (AK), ceftriaxone (CRO), cefuroxime (CFM), cefixime (CXM), ciprofloxacin (CIP), co-trimoxazole (SXT), gentamicin (CN), fosfomycin (FOS), imipenem (IMP), meropenem (MEM), nalidixic acid (NAL), nitrofurantoin (F), piperacillin-tazobactam (TZP), tetracycline (TE), tigecycline (TGC), tobramycin (TOB), colistin (CT), polymyxin (PB) and cefepime (FEP).

According to the standard CLSI guidelines, results were recorded by measuring the zone diameters and finally noted as sensitive (S), intermediate (I) or resistant (R). The isolates were categorized as multidrug-resistant when the isolates were found to be resistant to three or more antibiotics [24].

### 4.8. Antibiotic Panel for S. aureus

The following antibiotics (Thermo Fisher Scientific Inc., Washington, USA) were used for antibiotic susceptibility testing of *S. aureus*:

Amikacin (AK), cefoxitin (FOX), chloramphenicol (C), co-trimoxazole (SXT), ciprofloxacin (CIP), gentamicin (CN), linezolid (LZD), fusidic acid (FD), neomycin, norfloxacin (NOR), tetracycline (TE), penicillin (P), clindamycin (DA), erythromycin (E), tigecycline (TGC), teicoplanin (TEC), tobramycin (TOB), and vancomycin (VA).

All of the coagulase test positive *staphylococci* isolates were tested for resistance to FOX on MH Agar using 30 ug/mL FOX according to the standard guidelines of CLSI. As per CLSI guidelines, the resistant strains showed a zone diameter of <24 mm, while a diameter of >25 mm showed the susceptible (S) strains. Strains that were resistant to FOX were noted as MRSA [24].

#### 4.8.1. Screening of Methicillin and Vancomycin Resistance

The agar diffusion screening method was used for confirmation and screening of methicillin (MRSA/MSSA) and vancomycin resistance (VRSA/VSSA) [24].

#### 4.8.2. Determination of MIC

Strains of *S. aureus* have the ability to grow in MH Agar under aerobic conditions. To determine the minimum inhibitory concentrations (MICs) against vancomycin, E-Test strips (Biomeriux, Marcy-l’Etoile, France) were used. An MIC value of more than 16 µg/mL was considered as resistant to vancomycin on an MH agar plate comprising a 6 µg/mL vancomycin strip [24]. ATCC strain 29,213 was used as a control.

### 4.9. Antibiotic Panel for P. aeruginosa

The following antibiotics (Thermo Fisher Scientific Inc., Waltham, MA, USA) were used for antibiotic susceptibility testing of *P. aeruginosa*:

Amikacin (AK), ciprofloxacin (CIP), co-trimoxazole (SXT), gentamicin (CN), tobramycin (TOB), aztreonam (AZM), cefepime (FEP), ceftazidime (CAZ), colistin (CT), meropenem (MEM), imipenem (IPM), piperacillin-tazobactam (TZP) and polymyxin (PB) [24].

### 4.10. Molecular Identification of Bacterial Isolates

#### 4.10.1. DNA Isolation

The bacterial template DNA was isolated using a commercially available Wizard DNA extraction kit (Promega, Madison, Wisconsin, United States of America). The isolated bacterial colonies were inoculated in a sterile glass tube containing BHI culture medium (Oxoid, United States of America) and incubated overnight at 37 °C. After the incubation period, the overnight culture medium was transferred to the 1.5 mL microcentrifuge tube and centrifuged at 13,000 g for 2 min. Then, the supernatant was discarded, and the sediment was selected for further DNA extraction and purification using the manufacturer guidelines [11]. 

#### 4.10.2. Agarose Gel Electrophoresis

The quality of extracted DNA was verified using 1% agarose gel electrophoresis and visualized with the Gel Documentation System (Bio-Rad, CA, USA) [1].

#### 4.10.3. Identification of Bacterial Isolates through PCR with Different Primers

The primer sequences were obtained from the previous studies and evaluated for standard protocols. The list of primers for identification of MRSA and other bacterial isolates is shown in Table 4. The initial denaturation temperature for *fimh, usp, IMP* and *AIM* genes was 95 °C for 10 min. For *nuc* and *mecA* genes, the initial denaturation was performed at 95 °C for 5 min. Denaturation for *fimh, usp, IMP* and *AIM* genes was carried out at 95 °C for 30 s, for *nuc* at 94 °C for 15 s and for *mecA* gene at 94 °C for 20 s. The annealing temperature for each gene amplification is given in Table 4. The extension for *fimh, usp*, *nuc* and *mecA* gene was completed at 72 °C for 1 min, while for *IMP* and *AIM* genes it was at 72 °C for 50 s. The number of repeated cycles for *fimh* and *usp* was 34. For *IMP* and *AIM*, there were 36 and 30 *nuc* and *mecA* genes amplified. For all genes, the final elongation step was at 72 °C for 10 min.

### 4.11. Statistical Analysis

The data, including the prevalence of bacterial isolates, antibiotic susceptibility data (sensitivity and resistance) and molecular identification of genes (positive or negative), were transferred to a Microsoft Office Excel spreadsheet version 2016 (Microsoft, Washington, DC, USA). The data were analyzed for absolute values and reported in percentages.

## 5. Conclusions

The present study highlighted the role of antimicrobial resistance in the exclusive infection rate of various uropathogens. The uropathogens were identified from different clinical samples and amplified via PCR to evaluate the presence of antibiotic resistance and virulent genes. Further studies on a large scale are needed to investigate the resistance mechanism against various antibiotics.

## Figures and Tables

**Table 1 antibiotics-11-00516-t001:** List of collected samples and the prevalence of different bacterial infections.

Organisms	Urine	Foley Tip	Cystoscopy Urine	Nephrostomy Urine	Total Positive
*E. coli*	607	16	26	32	681
*P. aeruginosa*	95	08	15	24	142
*Klebsiella pneumoniae*	24	01	04	12	41
*Proteus mirabilis*	04	-	-	03	07
*Acinetobacter baumanni*	11	03	01	04	19
*Enterobacter cloacae*	03	-	01	02	06
*Citrobacter* spp.	05	-	-	01	06
*Staphylococcus aureus*	152	07	06	18	183
*Enterococcus faecium*	01	-	01	02	04
*Enterococcus faecalis*	13	01	01	03	18
Total organism	915	36	55	101	1107
Total no. of samples	1534	94	104	167	1899

**Table 2 antibiotics-11-00516-t002:** Antibiotic resistance pattern of *E. coli* and *P. aeruginosa*.

Antibiotics	Concentration (µg)	*E. coli* (*n* = 681)		*P. aeruginosa* (*n* = 142)	
		S (*n*)	R (*n*/%)	S (*n*)	R (*n*/%)
Ampicillin (AMP)	10	46	635 (93.24)	NT	NT
Amp-clavulanic acid (AMC)	20	76	605 (88.83)	NT	NT
Amikacin (AK)	30	598	83 (12.18)	126	16 (11.26)
Aztreonam (AZM)	30	NT	NT	24	118 (83.09)
Ceftriaxone (3G) (CRO)	30	131	550 (80.76)	NT	NT
Ceftazidime (CAZ)	30	NT	NT	107	35 (24.64)
Cefuroxime (2G) (CFM)	30	104	577 (84.72)	NT	NT
Cefixime (3G) (CXM)	5	104	577 (84.72)	NT	NT
Ciprofloxacin (CIP)	5	91	590 (86.63)	NT	NT
Co-trimoxazole (SXT)	23.75	138	543 (79.73)	NT	NT
Gentamicin (CN)	10	601	80 (11.74)	116	26 (18.30)
Fosfomycin (FOS)	200	609	72 (10.57)	NT	NT
Imipenem (IMP)	10	654	27 (3.96)	104	38 (26.76)
Meropenem (MEM)	10	654	27 (3.96)	84	58 (40.84)
Nalidixic acid (NAL)	30	36	645 (94.71)	NT	NT
Nitrofurantoin (F)	300	598	83 (12.18)	NT	NT
Piperacillin-tazobactam (TZP)	10	571	110 (16.15)	129	13 (9.15)
Tetracycline (TE)	30	117	564 (82.81)	NT	NT
Tobramycin (TOB)	10	634	47 (6.90)	98	44 (30.98)
Colistin (CT)	10	NT	NT	137	05 (3.52)
Polymyxin (PB)	300 units	NT	NT	136	06 (4.22)
Cefepime (FEP)	30	NT	NT	73	69 (48.59)

NT: not tested. S: sensitive. R: resistant.

**Table 3 antibiotics-11-00516-t003:** Antibiotic resistance pattern of *S. aureus* (*n* = 183).

Antibiotics	Concentration (µg)	Antibiotic Susceptibility Testing		Resistance %
		Sensitive	Resistant	
Amikacin (AK)	30	118	65	35.51
Cefoxitin (FOX)	30	152	31	16.93
Ciprofloxacin (CIP)	5	76	107	58.46
Co-trimoxazole (SXT)	23.75	89	94	51.36
Gentamicin (CN)	10	107	76	41.53
Linezolid (LZD)	30	183	00	00
Clindamycin (DA)	2	136	47	25.68
Erythromycin (E)	15	89	94	51.36
Nitrofurantoin (F)	300	179	04	2.18
Penicillin (P)	10 units	07	176	96.17
Tetracycline (TE)	30	29	154	84.15
Teicoplanin (TEC)	30	183	00	00
Tobramycin (TOB)	10	111	72	39.34
Vancomycin (VA)	MIC	183	00	00

MIC: minimum inhibitory concentration.

**Table 4 antibiotics-11-00516-t004:** Primer sequences of genes of *E. coli, P. aeruginosa and* MRSA.

Organisms	Genes	Sequences of Primer (5′-3′)	Annealing	Reference
*E. coli*	*fimH*	F-TGCAGAACGGATAAGCCGTGG	60 °C for 1 min	[5]
		R-GCAGTCACCTGCCCTCCGGTA		
	*usp*	F-ACATTCACGGCAAGCCTCAG	58 °C for 1 min	[5]
		R-AGCGAGTTCCTGGTGAAAGC		
*P. aeruginosa*	*IMP*	F-GAAGGCGTTTATGTTCATAC	55 °C for 1 min	[1]
		R-GTATGTTTCAAGAGTGATGC		
	*AIM*	F-CTGAAGGTGTACGGAAACAC	54 °C for 1 min	[1]
		R-GTTCGGCCACCTCGAATTG		
*S. aureus*	*nuc*	F-GCGATTGATGGTGATACGGTT	50 °C for 30 s	[34]
		R-AGCCAAGCCTTGACGAACTAAAG		
	*mecA*	F-GATCGCAACGTTCAATTTAATTT	50 °C for 30 s	[34]
		R-GCTTTGGTCTTTCTGCATTCCT		

## Data Availability

Data can be shared upon reasonable request to namalik288@gmail.com.

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
