# Peer review of "The Molecular Characterization of Virulence Determinants and Antibiotic Resistance Patterns in Human Bacterial Uropathogens"

_antibiotics, 2022, doi:10.3390/antibiotics11040516_

Round 1

Reviewer 1 Report

In the revised manuscript, the authors have addressed properly for most of the concerns which were raised for the original manuscript. However, one concern still remained unaddressed.

I raised the concern about the definition of UTIs in the original manuscript. In the cover letter for the revised manuscript, the authors mentioned the criteria for the diagnosis of asymptomatic bacteriuria by IDSA, which was stated to be described in section 4.2. However, the recently updated guideline from IDSA (https://doi.org/10.1093/cid/ciy1121), was not used as a reference.

As was described in the guideline, the presence of symptoms suggestive of UTIs, together with the colony count of >105 CFU/mL, should be the definition of UTI for voided urine, instead of the colony count alone. Furthermore, samples from indwelling catheters should have the colony count of >105 CFU/mL instead of 102CFU/mL.

The authors should clearly show in the Results section how much of the 1899 samples they collected from the suspected patients met the definition of UTI mentioned above, not the number of samples with the presence of "pus cells" (white blood cells?)  and bacteria. If the authors feel difficult to re-evaluate their patient cohort at this stage, the authors should state that they did not differentiate symptomatic UTIs from asymptomatic bacteriuria.

Author Response

Reviewer 1:

Comments and Suggestions for Authors

In the revised manuscript, the authors have addressed properly for most of the concerns which were raised for the original manuscript. However, one concern still remained unaddressed. I raised the concern about the definition of UTIs in the original manuscript. In the cover letter for the revised manuscript, the authors mentioned the criteria for the diagnosis of asymptomatic bacteriuria by IDSA, which was stated to be described in section 4.2. However, the recently updated guideline from IDSA (https://doi.org/10.1093/cid/ciy1121), was not used as a reference.

Response: (P6-7, L227-234) Thank you for your kind consideration. We have revised the manuscript accordingly, and the issue was addressed with changes as per your insightful suggestions and comments. Furthermore, the sentence about symptomatic and asymptomatic cases has been strengthen by the following citation:

  1. Nicolle, L.E.; Gupta, K.; Bradley, S.F.; Colgan, R.; DeMuri, G.P.; Drekonja, D.; Eckert, L.O.; Geerlings, S.E.; Köves, B.; Hooton, T.M.; et al. Clinical Practice Guideline for the Management of Asymptomatic Bacteriuria: 2019 Update by the Infectious Diseases Society of Americaa. Clinical Infectious Diseases 2019, 68, e83-e110, doi:10.1093/cid/ciy1121.

As was described in the guideline, the presence of symptoms suggestive of UTIs, together with the colony count of >105 CFU/mL, should be the definition of UTI for voided urine, instead of the colony count alone. Furthermore, samples from indwelling catheters should have the colony count of >105 CFU/mL instead of 102CFU/mL.

Response: (P7, L134) Corrected as suggested.

The authors should clearly show in the Results section how much of the 1899 samples they collected from the suspected patients met the definition of UTI mentioned above, not the number of samples with the presence of "pus cells" (white blood cells?)  and bacteria. If the authors feel difficult to re-evaluate their patient cohort at this stage, the authors should state that they did not differentiate symptomatic UTIs from asymptomatic bacteriuria.

Response: (P3, L101-102) Corrected as suggested.

Reviewer 2 Report

The Quality of the manuscript was significant increased. There are 3 suggestions for an improvement in the recent version:

Line 188: The sentence starting with "While" is not a complete sentence. I guess it should be connected to former sentence.

Line 250: Doublecheck - have you really tested coagulase on a filter paper?

Line 258: Have you really mixed the bacteria with McFarland standard? Why?

Line 268: Replace "thickness" - maybe "concentration"

Author Response

Reviewer 2:

Comments and Suggestions for Authors

The Quality of the manuscript was significant increased. There are 3 suggestions for an improvement in the recent version:

Response: Dear reviewer, we would like to appreciate your kind suggestions and comments on our manuscript. After addressing your comments, our manuscript become better for the readers.

Line 188: The sentence starting with "While" is not a complete sentence. I guess it should be connected to former sentence.

Response: (P6, L189) Corrected as suggested.

Line 250: Doublecheck - have you really tested coagulase on a filter paper?

Response: (P7, L257). The sentence has been corrected and the word “filter paper” has been replaced with “Disposable Reaction Cards.”

Line 258: Have you really mixed the bacteria with McFarland standard? Why?

Response: (P7, L265). The term “McFarland standard” has been replaced with “Sterile 5 ml suspension medium (0.85% NaCl).”

Line 268: Replace "thickness" - maybe "concentration"

Response: (P7, L275) Corrected as suggested.

This manuscript is a resubmission of an earlier submission. The following is a list of the peer review reports and author responses from that submission.

Round 1

Reviewer 1 Report

General comments:

In the manuscript “The molecular surveillance of virulence determinants and antibiotic resistance patterns in human bacterial uropathogens” many microbiological basics are described. The basics need to be deleted. Everything that is described in a basic student book need not to be mentioned here – one example: E. coli is gram negative in the results section.

Many methods are standard methods and an easy citation is the correct way – example: Gram staining or preparation of Mc Farland standards. Only methods which are self- developed or adjusted must be described in detail.

All abbreviations must be explained (completely written) at the 1st occurrence.

Supplier names must be given for all reagents – see agar or antibiotic discs.

There must be a space between numbers and units.

Some terms are slang: “checked” – replace with “investigated” or “tested for” or “evaluated for the presence of”.

Lab abbreviations are not scientific standard: Terms like “+ve” or “saline bottle” need to be replaced.

Was there any consideration of Staphylococcus argenteus which is not easy to distinguish from MRSA?

Many important citations are not present or the described methods are self-developed.

Specific comments:

The title must be adjusted as the title sounds like a global investigation - it is a local study of a specific hospital – this must be clarified.

Line 65: “The Pseudomonas aeruginosa bacteria…” – “The species Pseudomonas aeruginosa” or maybe only “Pseudomonas aeruginosa”

Line 93: “is a form of staphylococcal bacteria this is immune” – it’s not a form of staphylococcal bacteria but a resistant variant of S. aureus. Note the difference between resistance and immunity.

Line 103: define “our country”

Line 123: it must be 681 E. coli

Line 143: Capital letters for bacterial genus names

Line 145: I guess DNAse – the nuclease- enzyme? Was this done on plates as well – it should be mentioned (not described) in the methods section with a citation.

Section 2.6.1 is a method description only and need to be moved to the methods section.

Line 164: “…are showed in percentages (%) of resistance and the percentages (%) of sensitive to various antibiotics.” Rewrite, maybe “relative abundance”

Figure 1: This figure gives not any essential or additional information. This figure must be deleted.

Table 4: This belongs to the methods section. Are the methods self- developed – otherwise the citation must be given. If there are no adjustments compared to published methods the table must be deleted and the PRC must only be mentioned with the appropriate citation.

Section 2.8: What is the point of a statistical analysis? This report is about abundancies of AMR and its correct reported as absolute and relative abundance. Why this additional statistics?

Line 193: What means: “P. aeruginosa reasons.”

Line 193 = Line 197: There is a 100% repeat of some sentences.

Line 205: “One of the previous study” this must be plural

Line 207: “The critical significance” This is not existent.

Line 211: “2nd-prevalant” numbers below 10 or at least from 0-5 must be written as word.

Line 227 and 230: “they” must be replaced by – “the authors”

Line 228: ”Their targets” rewrite

Line 245: As the nuc gene codes for the thermostable nuclease which is only present in S. aureus and S. argenteus and the mec gene codes for the methicillin resistance, a MRSA is defined by nuc: pos and mec: pos. As this is not a result of your study it should be deleted.

Line 306 ff.: The concentrations of the antibiotics on the discs must be given. Maybe a table is good for visualization.

Line 334: “(Biomeriux Strips)” delete “strips”

Line 348 ff.: This must be rewritten or deleted and only mentioned with an appropriate citation.

Table 5: Make clear which genes belongs to which species.

Line 366: “The % of antibiotic susceptibility” must be rewritten.

Reviewer 2 Report

Comments to the authors:

  1. As the authors may know, isolation of bacteria from urine does not always mean the presence of urinary tract infections. Asymptomatic bacteriuria should be distinguished from symptomatic infections. The authors of this study did not present the way they made the diagnosis of UTIs in their patients. The definition of UTIs which the authors relied on should be presented in the Material and Method section, and they should assess their data again to select true UTIs.
  2. In line 53 and line 127, what does +ve mean?
  3. The detailed explanation from line 59 to line 99 seems to be unnecessary and should be removed from the manuscript.
  4. In line 102 and 103, the authors state that "UTIs are one of a major burden in our country". They should present the reference regarding this fact.
  5. In Table 2, for P. aeruginosa, the percentage of resistance for imipenem and meropenem was not adequately presented.
  6. For molecular identification, from line 171 to 175, the authors analyzed the expression of genes such as fimH, usp, AIM, and nuc. Some readers would wonder why the authors decided to check these genes. The authors should describe the background of their work in Introduction.
  7. A detailed description of PCR reactions for each gene, shown in Figure 1 and Table 4, gives little information to the readers and should be removed. If they are truly necessary, the brief description should be added in the Materials and Methods.